# Wideband RCS Reduction Based on Hybrid Checkerboard Metasurface

**DOI:** 10.3390/s23084054

**Published:** 2023-04-17

**Authors:** Zhiming Zhao, Xiaoping Li, Guoxiang Dong

**Affiliations:** School of Aerospace Science and Technology, Xidian University, Xi’an 710126, China; 18131110408@stu.xidian.edu.cn (Z.Z.);

**Keywords:** RCS reduction, metasurface, polarization converter

## Abstract

Traditional stealth technologies all have their problems such as high cost and large thickness. To solve the problems, we used novelty checkerboard metasurface in stealth technology. Checkerboard metasurface does not have as high conversion efficiency as radiation converters, but it has many advantages such as small thickness and low cost. So it is expected to overcome the problems of traditional stealth technologies. Unlike other checkerboard metasurfaces, we improved it further by using two types of polarization converter units to be arranged in turn to form a hybrid checkerboard metasurface. Because the checkerboard metasurface composed of one type of polarization converter units can have a relatively wide radar cross-section (RCS) reduction in bandwidth when two types of polarization converter units are arranged in turn to form a hybrid checkerboard metasurface and mutual compensation of the two polarization converter units can broaden RCS reduction band further. Therefore, by designing the metasurface to be independent from the polarization, the effect of RCS reduction can be insensitive to the polarization of the incoming EM waves. The experiment and simulation results showed the value of this proposed hybrid checkerboard metasurface for RCS reduction. Mutual compensation of the units is a new attempt in the field of checkerboard metasurfaces for stealth technology and proved to be effective.

## 1. Introduction

Stealth technology is an important research topic worldwide and an enormous amount of research effort goes into stealth technology in order to reduce radar cross-section (RCS). Traditional stealth technologies include shape stealth technology and absorbing material stealth technology. The shape stealth technology utilizes the shape of the aircraft to achieve interference so as to increase radar tracking errors. Its disadvantage is that the stealth function’s demand for shape is inconsistent with the aerodynamics’ demand for shape. The stealth technology of absorbing material uses the strong absorption characteristics of aircraft surface coating materials to electromagnetic (EM) waves to significantly reduce reflected waves; its disadvantages are high cost and large thickness [1,2,3,4,5,6].

Because traditional stealth technologies have their own disadvantages, in this article, we used artificial metasurface in stealth technology, which can be expected to overcome the disadvantages of traditional stealth technologies. Metasurface is a periodic structure that has emerged since the 21st century, it is composed of subwavelength units and has special electromagnetic properties that natural materials do not have. These properties mainly come from artificial special structures rather than the materials of it [7,8,9]. Researchers have found that metasurface has a strong ability to control the propagation path, amplitude, direction, polarization, phase, and spectrum of EM waves. Since Landy and other scholars at Boston University proposed it in 2008 [10], it has received extensive attention from scholars around the world. Because it has a strong ability to control the EM waves, the metasurface can be used for stealth technology. Compared with traditional stealth technologies, metasurface stealth technology has the advantages of wide bandwidth, small thickness, the possibility to conform it to any object shape while retaining its properties, low cost, ease to process, and so on. At present, the main types of metasurface used for stealth are phase gradient metasurface, checkerboard metasurface, absorbing metasurface, and so on. For example, in 2022, Andrey V. Sabluk used a frequency-selective high-Q metamaterial for the fabrication of a terahertz-to-infrared converter. The converter can achieve electromagnetic radiation absorption coefficient of 99.998% and radiation conversion efficiency over 94% [11]. For checkerboard metasurface, as it can achieve broadband RCS stealth and with polarization insensitivity, it is more and more used in stealth technology. The Checkerboard metasurface used two alternatingly arranged different types of units to reduce the RCS based on the principle of backscattering cancellation between the reflected waves of two different units. The backscattering cancellation happens when the reflection amplitudes of the two different units are the same and the reflection phase difference between the two different units is around 180°. Moreover, when the backscattering cancellation occurs, the vertically incident EM waves are reflected and present an abnormal scattering state, the reflected waves in the main lobe direction are greatly reduced, thereby reducing the RCS of the target. At first, when the checkerboard metasurface was proposed, it was composed of artificial magnetic conductor (AMC) units and perfect electric conductor (PEC) units. When the incident waves illuminated the checkerboard metasurface, the reflected waves were offset based on the backscattering cancellation effect of the contributions of the AMC units and PEC units. In 2007, a such kind of checkerboard metasurface was first proposed composed of AMC units and PEC units, the checkerboard metasurface can achieve monostatic RCSR of 10 dB at the frequency band ranging from 15 GHz to 16 GHz [12]. Then, some scholars improved the checkerboard metasurface’s stealth effect by changing the unit of PEC structure into the unit of AMC structure. The new checkerboard metasurface can be designed by using two different types of AMC unit structures [13] or one type of AMC unit structure with different sizes [14]. For example, a square structure and loop structure [15,16], a square structure and circular structure [13], a cross structure and square structure [17,18], or a cross structure with different sizes [19,20,21]. Compared with checkerboard metasurface based on AMC units and PEC units, the RCS reduction effect of checkerboard metasurface is based on two different types of AMC units or one type of AMC units with different sizes is better. This is because checkerboard metasurface based on the way can achieve a 180° phase difference in a wider frequency band, thus expanding the RCS reduction bandwidth. However, the way still has the disadvantage of lack of optimization methods for expanding the RCS reduction bandwidth. Because optimization methods only have optimized the units’ sizes of checkerboard metasurface so far, how to find a new method to expand the RCS reduction bandwidth is desired. Along with the development of metasurface technologies, researchers found that another kind of metasurface used in polarization converters can also be used in stealth checkerboard metasurfaces due to its own characteristics [22,23,24,25,26,27]. Researchers found that both the unit and mirror unit of a metasurface polarization converter can achieve 180° phase difference in a wider frequency band than the units of traditional checkerboard metasurface, the finding provides a basis for the design of checkerboard metasurfaces which are composed of the units of metasurface polarization converters. There have been related research using one type of metasurface polarization converter units in checkerboard metasurface design. For example, in 2021, Qi Zheng used MS units to compose a metasurface to achieve an RCS reduction of over 10 dB from 3 to 13 GHz [28]. In 2021, Changfeng Fu used a linear polarization conversion coding metasurface to achieve an RCS reduction of over 10 dB from 9.5 to 13.9 GHz and 15.2–20.4 GHz [29]. In 2022, Joysmita Chatterjee used a metasurface-based polarization converter for ultrawideband RCS reduction, it can achieve RCS reduction of over 10 dB from 5.5 to 20.5 GHz [30]. The above articles can prove that checkerboard metasurface based on polarization converter units can achieve broadband RCS reduction.

Unlike previous checkerboard metasurface composed of one type of units of metasurface polarization converter, in this paper, we proposed a hybrid checkerboard metasurface composed of two types of units of different metasurface polarization converters for wideband RCS reduction, and the two types of units were distributed in a certain way. At first, we designed the checkerboard metasurfaces based on one type of polarization converter unit and they both can realize relatively wideband RCS reduction. However, the RCS reduction bands of the checkerboard metasurfaces based on each type of polarization converter unit were located in different frequency bands. When these two types of polarization converter units were placed in turn to form the proposed hybrid checkerboard metasurface, the RCS reduction bands of the hybrid checkerboard metasurface were expanded further because of the compensation of the different frequency bands. The RCS of the hybrid checkerboard metasurface was obtained by simulation, and the simulation results showed that wideband RCS reduction was achieved based on this hybrid checkerboard metasurface. The −10 dB RCS reduction bands ranged from 13 GHz to 24 GHz and from 29.5 GHz to 33.7 GHz for x-polarized incidences, while the −10 dB RCS reduction bands ranged from 19.8 GHz to 27.0 GHz and from 29 GHz to 35.5 GHz for y-polarized incidences. Finally, we performed experimental verifications, and the results obtained were basically consistent with the simulation results. 

## 2. Designs and Methods

The checkerboard metasurface is composed of two different units arranged alternately. Generally, the metasurface includes many such units. In order to briefly introduce the structure of the checkerboard metasurface, the simplest form of the checkerboard metasurface model is proposed and it is shown in Figure 1, the checkerboard metasurface model includes four units, unit A_1_ and unit A_2_. If incident EM waves are reflected by the metasurface, in the working frequency band of the metasurface, the metasurface can achieve backscattering cancellation, in this time the reflected waves will be dispersed in different directions. As the reflection path of the reflected waves is changed and the reflected waves are in an abnormal scattering state, the reflected waves in the *z*-axis direction are greatly reduced, and the RCS of the metasurface can be greatly reduced together. Since the international general standard is that the RCS reduction value of the stealth structure and the PEC of the same size must reach more than 10 dB, the standard should be followed when designing the checkerboard metasurface. The RCS reduction in checkerboard metasurface and PEC with the same size is shown in Formula (1), where the electric field strength of the incident waves is *E*_i_ and the electric field strength of reflected waves is *E*_s_.
(1)$$\begin{array}{cc} \mathrm{RCS}\mathrm{reduction} & =10\log\left[\frac{\underset{r\to \infty }{\lim}\left[4{\pi r}^{2}\frac{{\left|E_{s}\right|}^{2}}{{\left|E_{i}\right|}^{2}}\right]}{\underset{r\to \infty }{\lim}\left[4{\pi r}^{2}{\left(1\right)}^{2}\right]}\right]\\ & =10\log\left[\frac{{\left|E_{s}\right|}^{2}}{{\left|E_{i}\right|}^{2}}\right] \end{array}$$

And *r* is the distance between the scattering field and the source of incident waves. For a scattering surface composed of two types of structure, because the area of unit A_1_ and unit A_2_ is equal, RCS reduction can be expressed as Formula (2): (2)$$\mathrm{RCS}\mathrm{reduction}=10\log{\left[\frac{A_{1}e^{{\mathrm{jP}}_{1}}+A_{2}e^{{\mathrm{jP}}_{2}}}{2}\right]}^{2}$$where *A*_1_ and *A*_2_ represent the reflection amplitudes of the unit A_1_ and unit A_2_, and *P*_1_ and *P*_2_ represent the reflection phases of the unit A_1_ and unit A_2_. When unit A_1_ and unit A_2_ can achieve total reflection, the reflection amplitudes *A*_1_ and *A*_2_ are the same, so Formula (2) can be simplified to Formula (3):(3)$$\mathrm{RCS}\mathrm{reduction}=10\log\frac{1+\cos(P_{1}-P_{2})}{2}$$

In order to realize 10 dB RCS reduction, Formula (3) is simplified in a series. Finally, we can acquire Formula (4), so the reflection phases of the two AMC units should satisfy Formula (4):(4)$$10\log\frac{1+\cos(P_{1}-P_{2})}{2}\leq -10\;\mathrm{dB}\Rightarrow 143{}^{\circ}\leq \left|P_{1}-P_{2}\right|\leq 217{}^{\circ}$$

It is worth noting that, in Formula (1), the ratio of incident electric field intensity and the scattering electric field intensity is only equal to the reflection coefficient of the checkerboard metasurface. Because the checkerboard metasurface consists of two different AMC units, each unit occupies almost half of the total space of the checkerboard metasurface. Thus, the reflection coefficient of the checkerboard metasurface can be approximately equal to the average value of the reflection coefficient of the two kinds of AMC units. Therefore, RCS reduction can be expressed as Formula (2) approximately. Although Formula (2) does not consider the edge effect, it provides good guidelines for the RCS reduction in a surface and the surface possesses two types of structures. Furthermore, it has been proven to be valid in many studies. Therefore, in order to obtain wideband RCS reduction, two different AMC units satisfying Formula (4) should be designed in wideband.

In this work, we utilized two different metasurface polarization converter units. For one of them, we designed a wideband high-efficiency polarization converter unit. The unit structure of the polarization converter is shown in Figure 2a, its geometric parameters are *p* = 4 mm, l = 3.8 mm, g = 1.2 mm, d = 0.2 mm, w = 0.2 mm, a = 1.7 mm, and b = 1.1 mm. The metal patch is loaded on the front of the dielectric substrate and the layer of metal is coated on the back of the dielectric substrate, the thickness of the dielectric substrate is 1.5 mm, the dielectric constant of it is 2.65, and the loss tangent is 0.001. For the other, we designed a perfect polarization converter unit. The unit structure of the polarization converter is shown in Figure 2b. Its geometric parameters are *p* = 3 mm, l = 2.9 mm, g = 0.45 mm, a = 0.73 mm, b = 2 mm, w = 0.2 mm, r1 = 1.3 mm, and r2 = 0.9 mm. Its dielectric substrate is the same as the one of the wideband high-efficiency polarization converter. The full-wave simulation has been performed by the simulation software Computer Simulation Technology (CST). The reflection characteristics of the wideband high-efficiency polarization converter unit and its mirror structure are shown in Figure 3. The rotation angle φ=45° represents the high-efficiency polarization converter unit, and the rotation angle φ=−45° represents the mirror structure of the high-efficiency polarization converter unit. It is shown in Figure 3 that the polarization converter unit and its mirror structure can both achieve wideband cross-polarization conversions with high efficiency. This enables the reflection phase difference between the polarization converter unit and its mirror structure to satisfy Formula (4) in a very wide frequency band which is from 24 GHz to 37.5 GHz, and the reflection amplitudes of them were both almost 1 in this band. Therefore, the wideband high-efficiency polarization converter unit and its mirror structure are expected to compose a checkerboard metasurface for RCS reduction, the RCS reduction band is from 27 GHz to 37.5 GHz. Similarly, Figure 4 shows the reflection characteristics of the perfect polarization converter unit and its mirror structure. The angle φ=45° represents the perfect polarization converter unit, and the angle φ=−45° represents the mirror structure of the perfect polarization converter unit. As shown in Figure 4, the perfect polarization converter unit and its mirror structure can enable reflection phase difference between the polarization converter unit and its mirror structure to satisfy the Formula (4) in a very wide frequency band which is from 21.5 GHz to 34 GHz, and the reflection amplitudes of them are both almost 1 in this band. Therefore, the perfect polarization converter unit and its mirror structure are expected to compose a checkerboard metasurface for RCS reduction, the RCS reduction band is from 21.5 GHz and 34 GHz.

Because the expected −10 dB RCS reduction bands of the two kinds of checkerboard metasurfaces do not coincide, in order to achieve better RCS reduction, we propose a hybrid checkerboard metasurface based on the two kinds of polarization converter units. The hybrid checkerboard metasurface is composed of wideband high-efficiency polarization converter units for the lower band and perfect polarization converter units for the higher band. When the two types of polarization converter units are placed in turn to form the proposed hybrid checkerboard metasurface and the RCS reduction bands of the hybrid checkerboard metasurface are expanded further because of the compensation of the different frequency bands. 

As shown in Figure 5a, the two types of polarization converters are arranged in turn to compose the hybrid checkerboard metasurface, its overall size is 210 mm × 240 mm. The hybrid checkerboard metasurface is composed of base arrays in the form of 5 × 5. The base array is shown in Figure 5b, its overall size is 42 mm × 48 mm. The simulated monostatic RCS of the proposed hybrid checkerboard metasurface and PEC under different polarized waves are, respectively, shown in Figure 6 and Figure 7. For x-polarized vertical incidence EM waves, the −10 dB RCS reduction bands range from 15.8 GHz to 19.3 GHz and 21.4 GHz to 36.5 GHz. For y-polarized vertical incidence EM waves, the −10 dB RCS reduction bands range from 15.8 GHz to 19.4 GHz and 21.6 GHz to 36.6 GHz. Since the new type of checkerboard metasurface has excellent RCS reduction effects on EM waves in two orthogonal polarization directions, according to the vector characteristics of EM wave polarization, it can be concluded that this structure can maintain excellent RCS reduction effects on EM waves with different polarization directions. 

When EM waves are obliquely incident at an angle of 30°, the simulated RCS of the proposed hybrid checkerboard metasurface, and PEC under different polarized waves are shown in Figure 8 and Figure 9. For x-polarized oblique incidence EM waves, the −10 dB RCS reduction bands range from 15.9 GHz to 18.2 GHz and 20.7 GHz to 36.5 GHz. For y-polarized oblique incidence EM waves, the −10 dB RCS reduction bands range from 16 GHz to 18.5 GHz and 21.6 GHz to 32 GHz.

## 3. Results of Simulation and Measurement

The 3D bistatic RCS patterns are investigated to further verify the performance of the designed checkerboard metasurface based on hybrid polarization converters. For x-polarized vertical incidences, the 3D bistatic RCS pattern of the checkerboard metasurface at 16.9 GHz is shown in Figure 10a. The 3-D bistatic RCS pattern of PEC with the same size as the checkerboard metasurface at 16.9 GHz is shown in Figure 10b. It can be shown that the reflection direction of EM waves is changed after EM waves are reflected by the metasurface and EM waves present an abnormal scattering state, in this time EM waves spread out in different directions instead of vertical reflection as the metasurface has backscattering cancellation effect on EM waves. In contrast, it can be shown that EM waves are reflected vertically by PEC. In this way, the RCS of the metasurface is reduced. The bistatic RCS along the principal plane is shown in Figure 11. In the Phi = 0° principal plane, the maxima of the bistatic RCS of the proposed hybrid checkerboard metasurface is 28.94 dB less than that of the PEC, and in the Phi = 90° principal plane. The maxima of the bistatic RCS of the proposed checkerboard metasurface is 28.85 dB less than that of the PEC. Because the proposed hybrid metasurface and the PEC are both rectangles, the bistatic RCS of the PEC along the Phi = 0° principal plane and the Phi = 90° principal plane is different. The bistatic RCS along the diagonal plane is shown in Figure 12. In the diagonal plane, the maxima of the bistatic RCS of the proposed hybrid checkerboard metasurface is 28.94 dB less than that of the PEC. The bistatic RCS patterns show that the reflected fields are redirected by this proposed checkerboard metasurface compared with that of the PEC.

For x-polarized oblique incidence and the incident angle is 30°, the 3D bistatic RCS pattern of the proposed hybrid checkerboard metasurface is shown in Figure 13a; and the 3D bistatic RCS pattern of the PEC with the same size as the checkerboard metasurface is shown in Figure 13b. It can be seen that the metasurface still has a backscattering cancellation effect on EM waves. The reflection direction of EM waves is changed after being reflected by the metasurface and EM waves present an abnormal scattering state, in this time EM waves spread out in different directions instead of reflecting in one direction as the PEC. 

The 3D bistatic RCS patterns of the proposed checkerboard metasurface are also investigated under y-polarized vertical incidences. The 3D bistatic RCS pattern of the checkerboard metasurface is shown in Figure 14a and the 3-D bistatic RCS pattern of PEC with the same size is shown in Figure 14b at 17 GHz. Similar to the x-polarized EM waves, EM waves spread out in different directions instead of vertical reflection. The bistatic RCS along the principal planes and diagonal planes are, respectively, shown in Figure 15 and Figure 16. It can be shown that the RCS of the checkerboard metasurface along the principal plane is dramatically reduced. The maxima of the RCS of the proposed hybrid checkerboard metasurface are 23.53 dB less than that of the PEC in the Phi = 0° principal plane, and the maxima of the RCS of the proposed hybrid checkerboard metasurface is 23.57 dB less than that of the PEC in the Phi = 90° principal plane. In the diagonal plane, the maxima of the bistatic RCS of the proposed hybrid checkerboard metasurface is 23.53 dB less than that of the PEC.

For y-polarized oblique incidences and the incident angle is 30°, the 3D bistatic RCS patterns of the proposed hybrid checkerboard metasurface are investigated which is shown in Figure 17a, and the 3D bistatic RCS pattern of the PEC with the same size are investigated which is shown in Figure 17b. Similar to the x-polarized EM waves, the reflection direction of EM waves is changed after being reflected by the metasurface and EM waves present an abnormal scattering state. This time, EM waves spread out in different directions instead of reflecting in one direction as the PEC. In addition, mirror-backscattered lobes appeared, and this is because of the deterioration of the reflection phase of the unit of the proposed metasurface for oblique incidences.

In order to further verify the RCS reduction effect of the proposed hybrid checkerboard metasurface, the proposed actual sample of the hybrid checkerboard metasurface was tested through experiments. The actual sample of the proposed hybrid metasurface was handed over to a professional metasurface manufacturing company for processing and fabricating, and the sample is fabricated with the same parameters as the metasurface model used in the simulation. The measurement is carried out in the microwave anechoic chamber of our school. The actual metasurface sample is shown in Figure 18a, and a basic array of the actual metasurface sample is shown in Figure 18b. In the experimental process, the transmitting antenna and receiving antenna were located at the same height as the metasurface sample, and the actual metasurface sample was located vertically on a platform. In order to accurately calculate the RCS of the metasurface sample, the transmitting and receiving antennas were placed at a sufficient distance from the metasurface sample, and the antennas illuminated EM waves normally on the metasurface sample. The transmitting antenna and receiving antenna were both connected to an E8363B Agilent vector network analyzer, and the measured reflectivity was used to evaluate the RCS of the metasurface sample. As illustrated in Figure 19, the measured RCS of the metasurface sample is obtained for x-polarized vertical incidences, and as illustrated in Figure 20, the measured RCS of the metasurface sample is obtained for y-polarized vertical incidences. The measured results and the simulation results have good accordance, verifying the validity of the design of the proposed metasurface. There was a small difference between the measured results and the simulated results, which could be caused by a manufacturing error. The value of our proposed checkerboard metasurface for stealth technology was verified by the experiment.

As shown in Table 1, we compare the −10 dB bandwidth of this work with other works. It can be shown that the −10 dB bandwidth of the hybrid checkerboard metasurface in this work is the widest, and the hybrid checkerboard can achieve the RCS reduction effect for both x-polarized and y-polarized incident waves.

## 4. Conclusions

In this work, we proposed a hybrid checkerboard metasurface for RCS reduction. This hybrid checkerboard metasurface was composed of two types of polarization converter units. Compared with previous studies on RCS reduction, wideband RCS reduction was achieved by mutual compensation of the bands based on single types of polarization converter units. Both the experiment and simulation results showed that this hybrid checkerboard metasurface is a new application in stealth technology. The hybrid checkerboard metasurface does not have as high conversion efficiency as radiation converters, but it still has advantages, such as it can achieve wide bandwidth RCS reduction and the thickness of it is small. Therefore, it has a good prospect in the field of stealth.

## Figures and Tables

**Figure 1 sensors-23-04054-f001:**
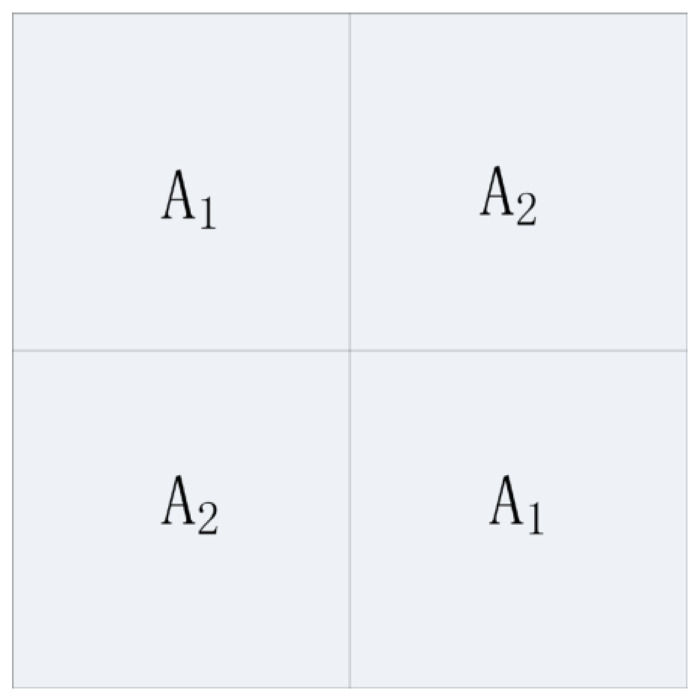
Schematic model used for analyzing the checkerboard metasurface.

**Figure 2 sensors-23-04054-f002:**
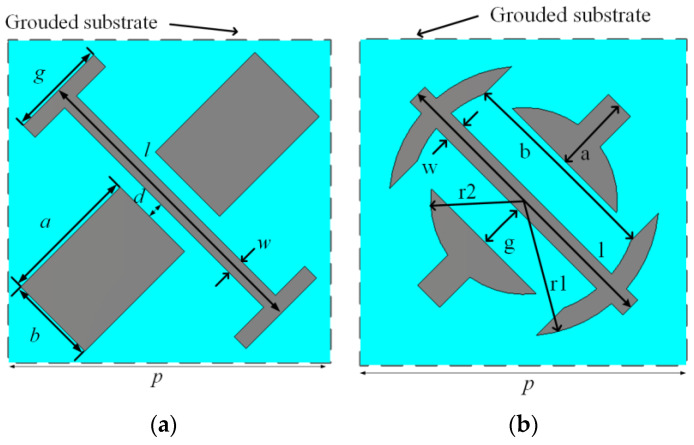
(**a**) Wideband high-efficiency polarization converter unit and (**b**) perfect polarization converter unit.

**Figure 3 sensors-23-04054-f003:**
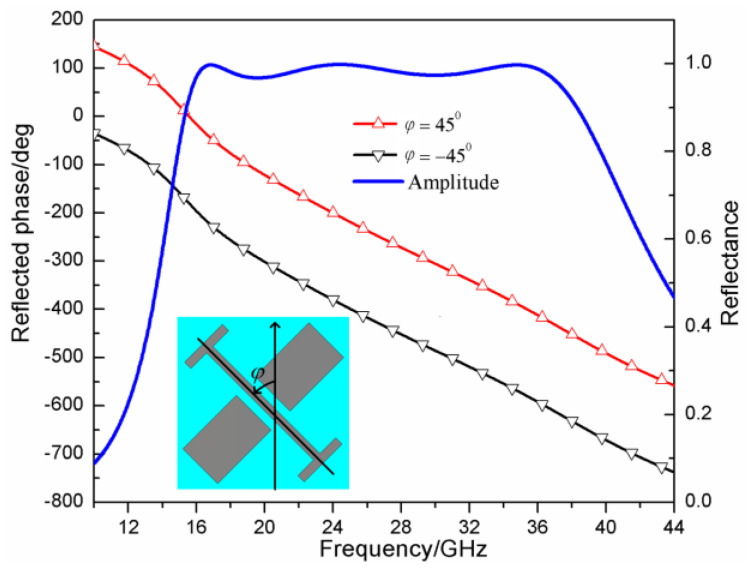
The reflection amplitude and phase of the wideband high−efficiency polarization converter unit as shown in the inner figure and its mirror structure, *φ* represents its rotation angle.

**Figure 4 sensors-23-04054-f004:**
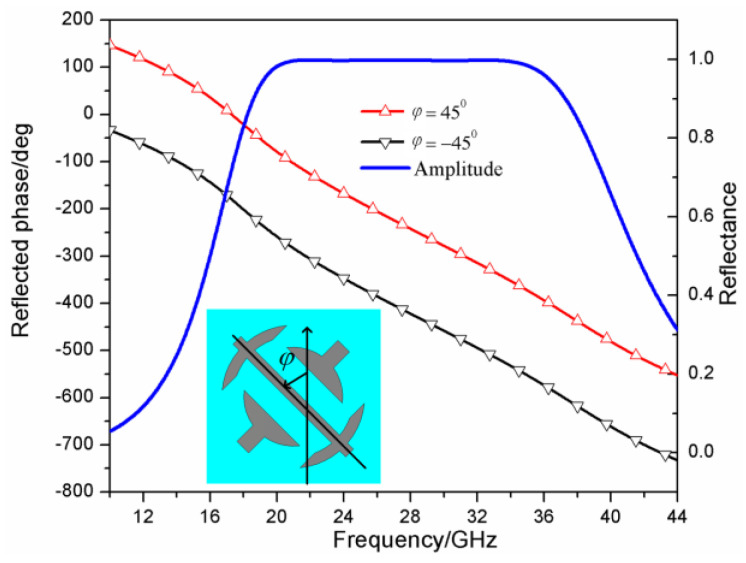
The reflection amplitude and phase of the perfect polarization converter unit as shown in the inner figure and its mirror structure, *φ* represents its rotation angle.

**Figure 5 sensors-23-04054-f005:**
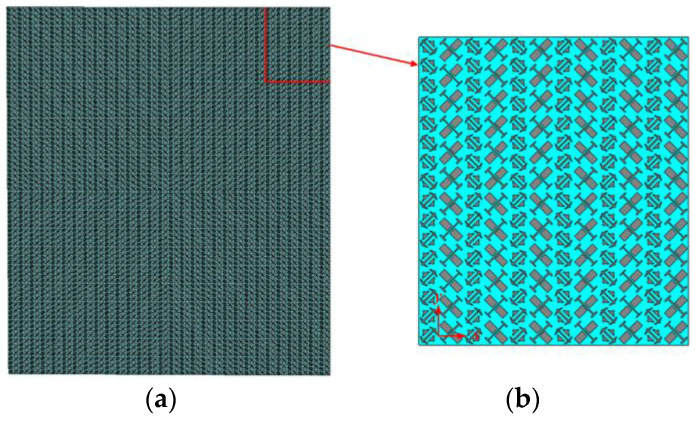
(**a**) The proposed hybrid checkerboard metasurface; (**b**) A basic array of the proposed hybrid checkerboard metasurface.

**Figure 6 sensors-23-04054-f006:**
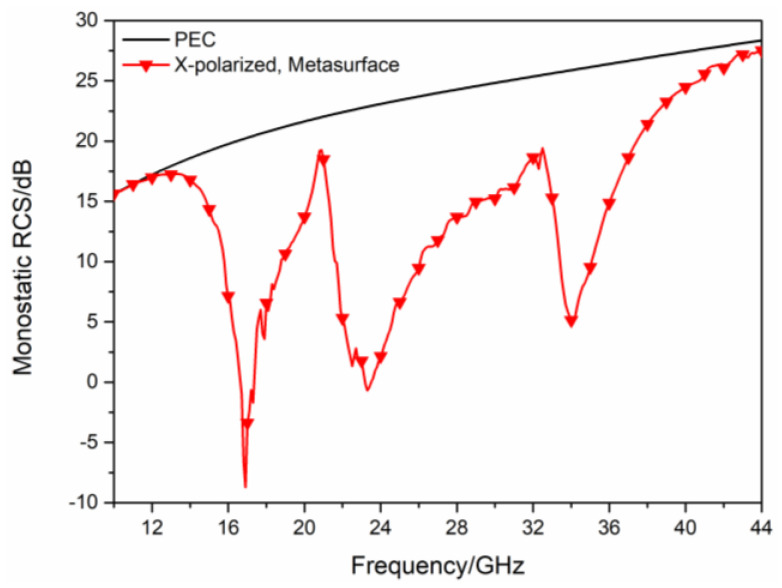
The RCS of the proposed hybrid checkerboard metasurface and PEC for x−polarized vertical incidence.

**Figure 7 sensors-23-04054-f007:**
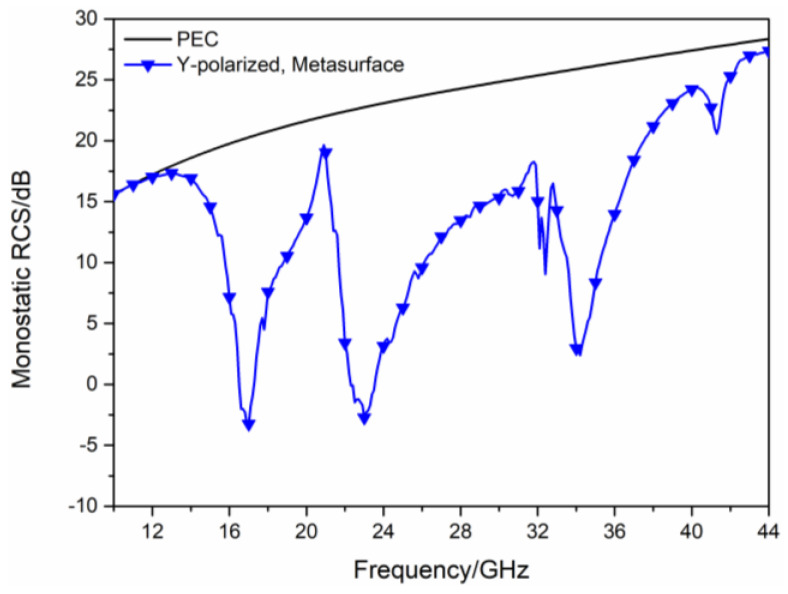
The RCS of the proposed hybrid checkerboard metasurface and PEC for y−polarized vertical incidence.

**Figure 8 sensors-23-04054-f008:**
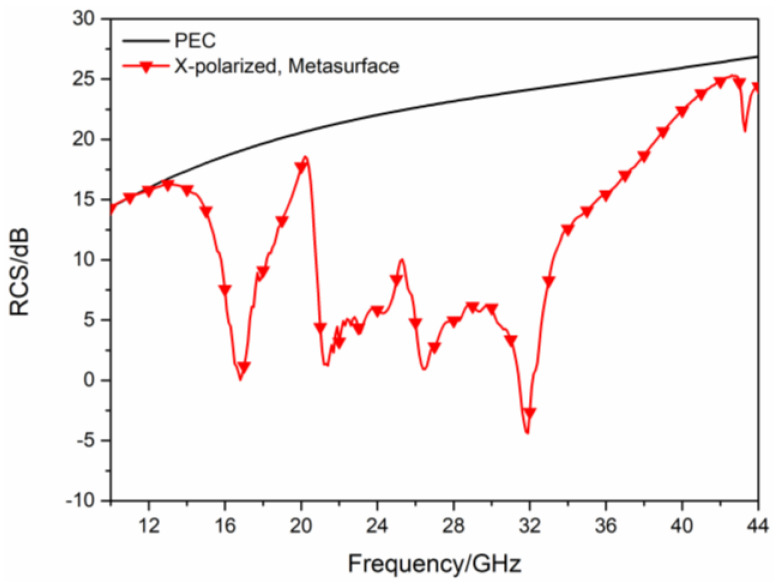
The RCS of the proposed hybrid checkerboard metasurface and PEC for x-polarized 30° oblique incidence.

**Figure 9 sensors-23-04054-f009:**
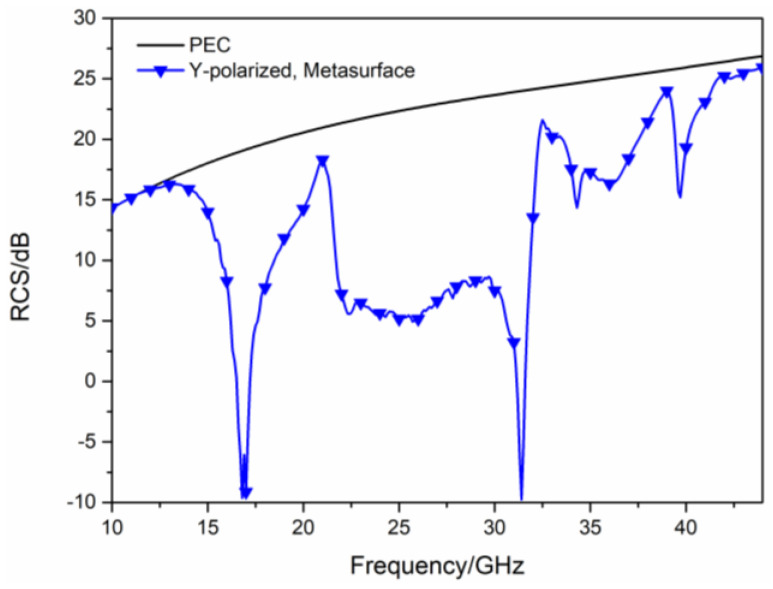
The RCS of the proposed hybrid checkerboard metasurface and PEC for y-polarized 30° oblique incidence.

**Figure 10 sensors-23-04054-f010:**
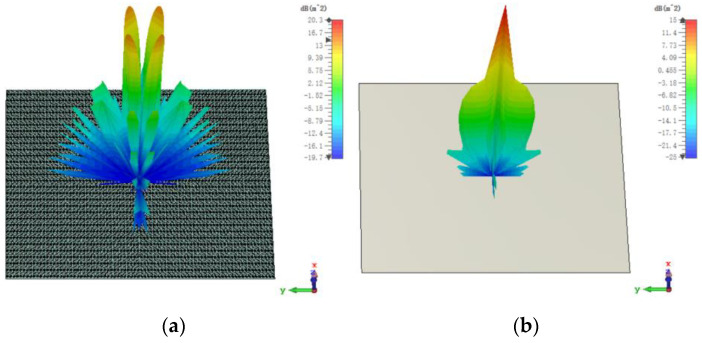
The 3-D bistatic RCS pattern of (**a**) checkerboard metasurface and (**b**) PEC at 16.9 GHz for x-polarized vertical incidence.

**Figure 11 sensors-23-04054-f011:**
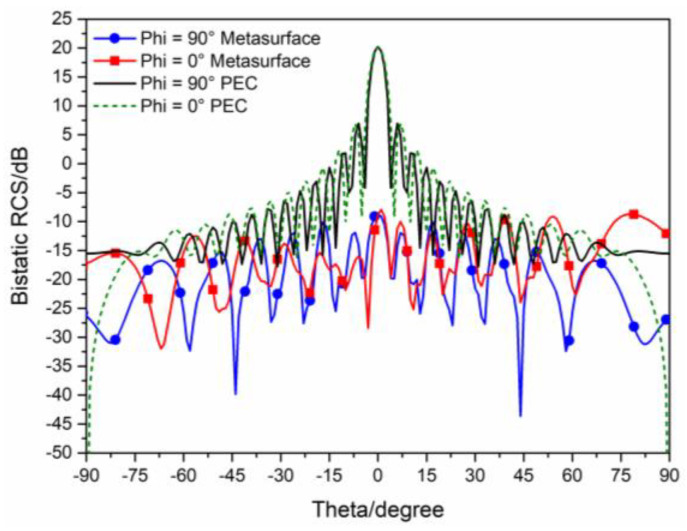
The bistatic RCS along principal planes at 16.9 GHz for x−polarized vertical incidence.

**Figure 12 sensors-23-04054-f012:**
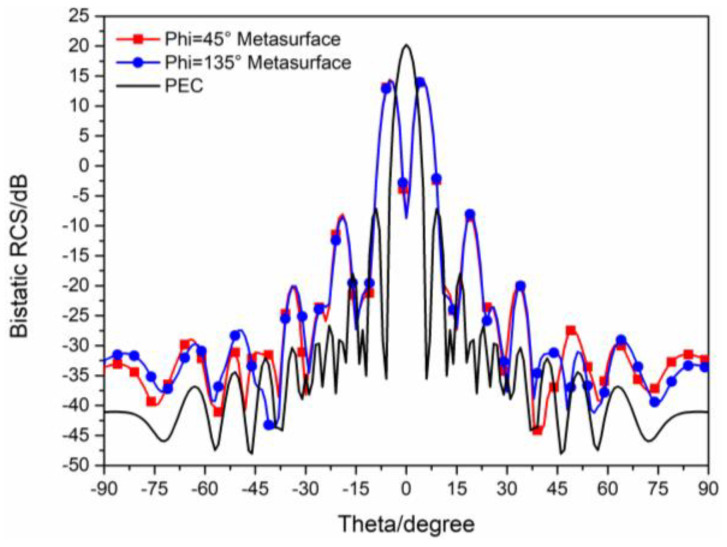
The bistatic RCS along diagonal planes at 16.9 GHz for x−polarized vertical incidence.

**Figure 13 sensors-23-04054-f013:**
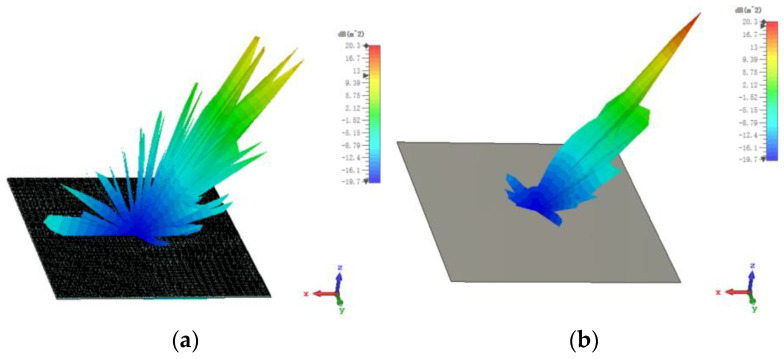
The 3-D bistatic RCS pattern of (**a**) checkerboard metasurface and (**b**) PEC at 16.9 GHz for x-polarized oblique incidence at an angle of 30°.

**Figure 14 sensors-23-04054-f014:**
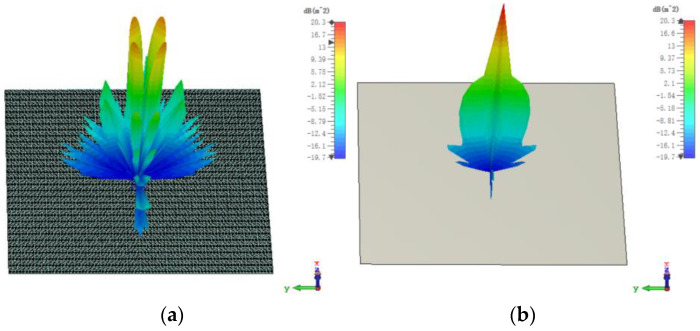
The 3-D bistatic RCS pattern of (**a**) checkerboard metasurface and (**b**) PEC at 17 GHz for y-polarized vertical incidence.

**Figure 15 sensors-23-04054-f015:**
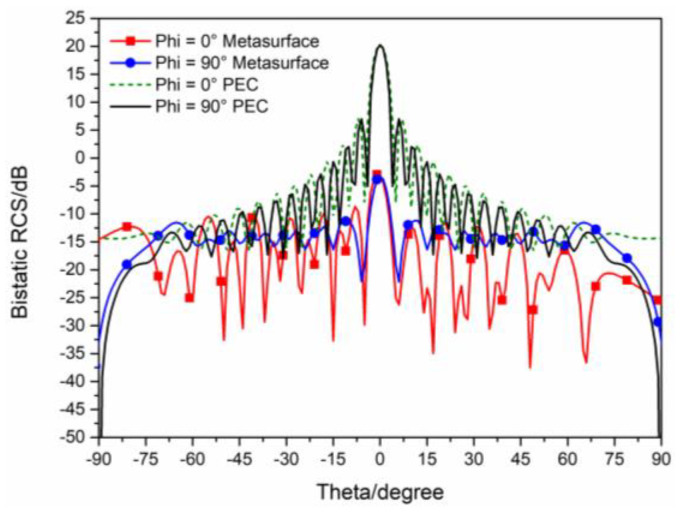
The bistatic RCS along principal planes at 17 GHz for y−polarized vertical incidence.

**Figure 16 sensors-23-04054-f016:**
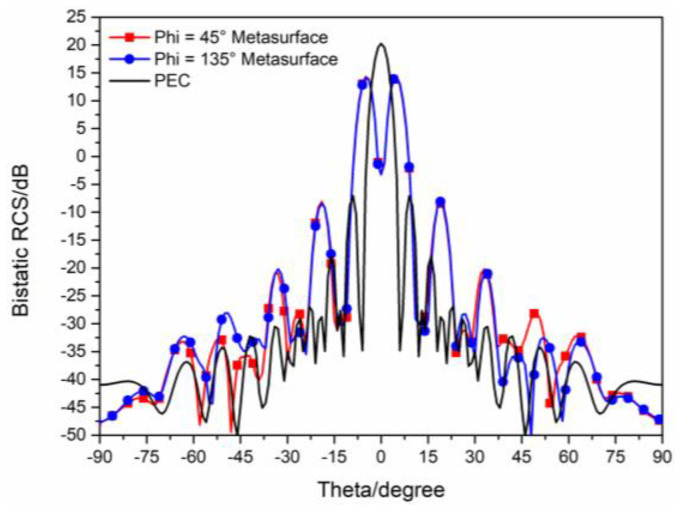
The bistatic RCS along diagonal planes at 17 GHz for y−polarized vertical incidence.

**Figure 17 sensors-23-04054-f017:**
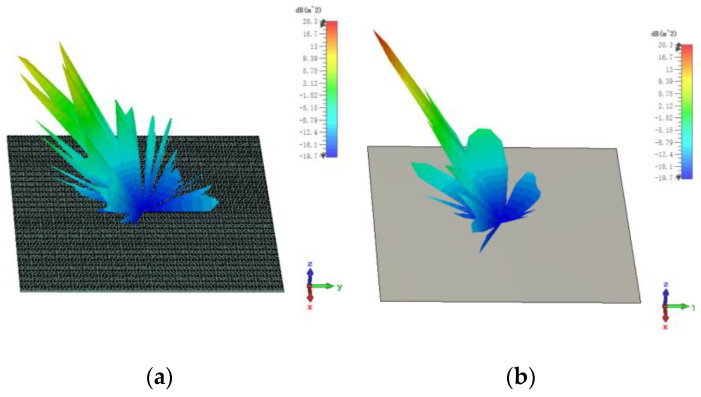
The 3-D bistatic RCS pattern of (**a**) checkerboard metasurface and (**b**) PEC at 17 GHz for y-polarized oblique incidence at an angle of 30°.

**Figure 18 sensors-23-04054-f018:**
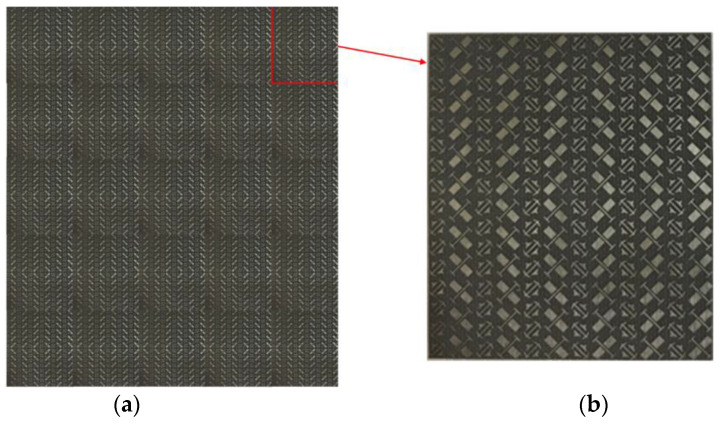
(**a**) Actual metasurface sample; (**b**) A basic array of the actual metasurface sample.

**Figure 19 sensors-23-04054-f019:**
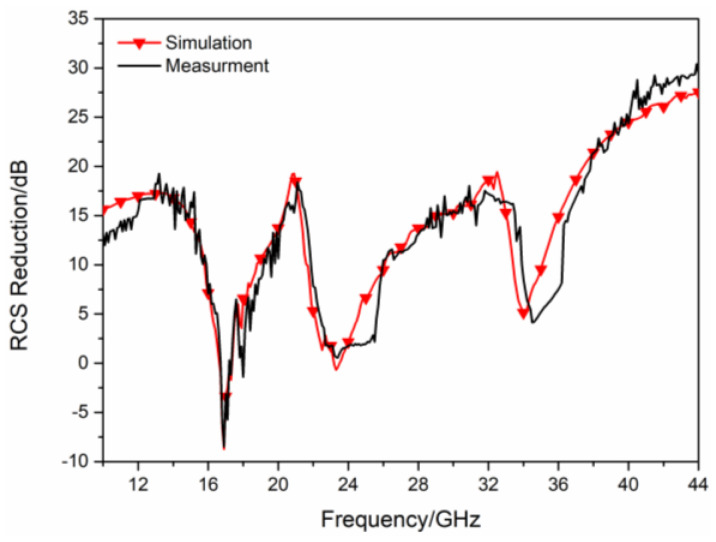
Measured RCS for x−polarized vertical incidence.

**Figure 20 sensors-23-04054-f020:**
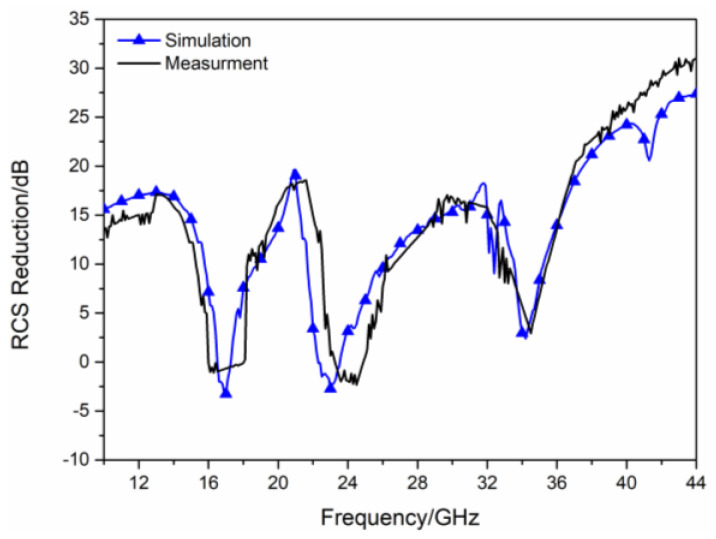
Measured RCS for y−polarized vertical incidence.

**Table 1 sensors-23-04054-t001:** Work band comparison.

Article	−10 dB Bandwidth (GHz)	Incidence Polarization
[28]	3–13	x,y
[29]	9.5–13.9; 15.2–20.4	x,y
[30]	5.5–20.5	x
This work	15.8–19.3; 21.4–36.5	x,y

## Data Availability

Not applicable.

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
