# Peer review of "Wideband RCS Reduction Based on Hybrid Checkerboard Metasurface"

_sensors, 2023, doi:10.3390/s23084054_

Round 1

Reviewer 1 Report (Previous Reviewer 1)

Authors proposed, simulated, fabricated and experimentally studied metasurface that can be interesting for further development of stealth technology. Obtained results are satisfactory for normal incidence, but the beampattern for oblique directions is not so good. So the design can be viewed as interesting step towards the development of high quality antireflection coating. Actually for practical cases of RCS reduction it is important not only reduce the back reflection, but also reduce back scattering and increase the absorption.  Presented figs 6, 7, 17, 18 for normal incidence are not so interesting for practical applications, in reality it is always oblique incidence.  The meaning of RSC=15-30 db for perfect electric conductor (PEC) in fig.6,7 is unclear. The perfect electric conductor completely reflects incident radiation back and its RSC=0 dB.  Fig.10 is also confusing: RSC varies from -40 dB up to +20 db, the last means that cross section is not reduced, but instead amplified by 20 dB.  It is not clear from description in p.4 lines 139-150 what is material for metal patch layer and back side of dielectric substrate.  One can assume that it is copper, that is good conductor and can be modelled in CST as PEC. Actually more efficient are radiation converters that absorb incident microwave radiation, convert it into infrared and scatter around, as in [A.Sabluk, A.Basharin, Metamaterial-based terahertz converter, Modern Electronic Materials 2022, 8(4), 149  DOI 10.3897/j.moem.8.4.98919] with radiation conversion efficiency over 94%. 

In the abstract and conclusions it would be useful for readers to see the frequency band and achieved RCS values. The drawbacks of the design should be also mentioned in abstract and conclusions.

Author Response

Reviewer 2 Report (New Reviewer)

1- The designs presented in recent years (especially from the papers of 2021-2023) should be reviewed. The most recent reference investigated in the manuscript is from 2017!

2- Formulas should be checked. How is formula 4 obtained?

3- The results are for the normal incidence. What behavior will result for the oblique incidence?

4- Performance characteristics of the proposed metasurface should be compared completely with the scientific reports (papers) presented in recent years through some appropriate figures and tables.

5- Figures 5 and 16 are not clear and are not useful with this low quality.

6- Figures 8, 11, 12, and 15 are not helpful.

7- The text of the manuscript and its content organization should to be revised accurately.

Author Response

Reviewer 3 Report (New Reviewer)

In the current research paper, the authors have used novelty of hybrid checkerboard metasurface for existing stealth technology for RCS reduction by using two types of polarization converter units.  The authors have ensured mutual compensation of the two polarization converter units which broaden the RCS reduction band. The experiment and simulation results showed the value of the proposed hybrid checkerboard metasurface for RCS reduction. Definitely mutual compensation of the units is a new attempt in the field of checkerboard metasurface for stealth technology and proved to be effective. The paper is duly supported by the mathematical equations wherever required and graphical representation and diagrams are also up to mark. The Paper is overall technically sound and should be considered for acceptance.

Author Response

This manuscript is a resubmission of an earlier submission. The following is a list of the peer review reports and author responses from that submission.

Round 1

Reviewer 1 Report

Manuscript is related to the development of stealth technology by reducing radar cross section.  Authors propose a novel hybrid checkerboard metasurface composed by two types of polarization converters previously studied in their publications [17, 18].  In experiment they obtained -10 dB reducing cross section in the frequency range 13-22 GHz.  Authors present in detail results of their numerical simulation using HFSS software for both types of polarization converters and for proposed hybrid. 

Some of results are not clear described and require some comments and clarification.  In Fig.6, 8, 10, 11 curves for metasurface vary from 0 down to -30 dB that is understandable, but smooth curves marked as PEC is a big puzzle. If it is perfect electric conductor it should just completely reflect RSCB=0 dB. What means value of +3 dB, does it mean that reflection area is twice as much as actual area? Why for PEC at lower frequencies below 12 GHz the reducing cross section is -10dB? Infinite PEC should completely reflect at all frequencies. If authors compare metasurface cross section with perfect electric conductor (PEC), the obvious illustration is to present the difference between signal reflected from PEC and metasurface.  It is not clear from text on page 5 what were the boundary conditions for simulations with HFSS, if it is the same as in experiment with 24x21 mm size, it is definitely too small because is comparable to the linewidth (21 mm is wavelength at 15 GHz).  It looks as if fig 3 and Fig.4 were calculated for infinite array with periodic boundary conditions.  For correct evaluation of such metasurface performance its dimensions should be 10 at least times as large.  And this is clear visible in RCS curves in Fig.3,4,6,810,11 where reflection even from PEC is about -10 dB, and from metasurfaces it is very nonuniform.  At frequencies over 40 GHz curves become much more smooth (wavelength below 0.7 cm that is 3 times smaller compared to sample size).  Beampatterns in Fig. 12-19 does not bring any valuable information for the same reason, sharp minima and sidelobes are mainly determined by small size of sample. 

Reviewer 2 Report

In this manuscript, the authors proposed a hybrid checkerboard metasurface for RCS reduction, which consists of two types of polarization converters. It is claimed that it can achieve wideband RCS reduction through both simulation and measurement. However, the manuscript is poorly written, and it cannot be accepted until the following comments are addressed properly.

1. There are multiple errors/typos across the paper. For instance, In Line 24, "illuminate" should be "illuminates". In Line 44, "design" should be "designed". In Line 95, "validated" should be "valid". In Line 161, "ideal" should be "idea". In Line 237, "be" should be deleted. In Line 274, "plan" should be "plane". There are other errors at other places. The authors should fix all of them before resubmitting.

2. The caption of Fig.2 is cut off. The text after (b) is missing.

3. The authors should reorganized the Introduction part. Right now it is not presented in a clear way and it is a bit duplicated and confusing.

4. The authors should provide more details on the fabrication of their prototype. Without this information, it is hard to judge the measurement data.

Reviewer 3 Report

I have several suggestions. First, in the introduction, make sure that you clearly state the prior art. For instance, use language such as "Prior research on FSS for RSC reduction used ...". Second, the English can be improved. Third, Figure 1 is not explained very well. Please improve the usefulness of Figure 1 and how it provides insight into your Hybrid FSS approach. Fourth, please provide a bit more explanation on why polarization performance of the FSS is so important. Some readers may not understand the reason you are concerned with polarization performance. Fifth, for the proposed approach, RCS in off principle axis directions is high (see Figures 12 and 16). I did not notice a comparison to the RCS patterns for the PEC case or in off normal incidence of the source. Please consider these suggestions. 

Round 2

Reviewer 1 Report

The second version of manuscript contains some clarification of the presented research including 3-D beampatterns of metasurface.  But author’s comments do not answer to questions from the first review.  All explanations are based on equation (1) that is correct, but not related to the actual size of the sample under test.  If the task of the work is to develop stealth technology for practical applications, this metasurface should cover large area of the order of square meters, not 48x48 or 24x21 mm for the wavelengths 0.7 - 3 cm.  As a result, the actual performance of stealth should be quite different from presented figures.  Authors correctly mention in their cover letter that RCS for PEC is exactly 0 dB, contrary to their figs. 6,8,10,11 with RCS from -12 to+3 dB.  Still no answer to the question: what means value of +3 dB, does it mean that reflection area is twice as much as actual area? The sample dimensions 24x21 mm is definitely too small because is comparable to the linewidth (21 mm is wavelength at 15 GHz).  Once again: for correct evaluation of stealth metasurface performance its dimensions should be at least 8 times as large.  At frequencies over 40 GHz curves become much more smooth (wavelength below 0.7 cm that is 3 times smaller compared to sample size).  Beampatterns in Fig. 12-19 does not bring any valuable information for the same reason, sharp minima and sidelobes are mainly determined by small size of sample. 

The article entitled “wideband RCS reduction” intended for development of stealth technology should contain the reflection reduction for the area much larger compared to the wavelength.  Both CST simulation and experimental studies should be done for the area 10 times as large compared to lowest frequency. If it is 10 GHz, the area should be 30x30 cm, for 30 GHz over 10x10cm, for smaller dimensions spectral characteristics are much dependent on the outer dimension of sample. No references for practical examples of stealth technology in the same wavelength that are in use for aircrafts defense. 

Reviewer 2 Report

Thanks for your responses to my comments. As for additional comments, I have to say even after the revision, the paper is still poorly written without sound logic or language. The scientific contribution of this work is low. Therefore, I don't recommend its publication.